

# The complete mitochondrial genome of *Lerema accius* and its phylogenetic implications

Qian Cong[1] and Nick V. Grishin[1,2]

[1] Departments of Biophysics and Biochemistry, University of Texas Southwestern Medical Center, Dallas, TX, United States
[2] Howard Hughes Medical Center, University of Texas Southwestern Medical Center, Dallas, TX, United States

## ABSTRACT

Butterflies and moths (Lepidoptera) are becoming model organisms for genetics and evolutionary biology. Decoding the Lepidoptera genomes, both nuclear and mitochondrial, is an essential step in these studies. Here we describe a protocol to assemble mitogenomes from Next Generation Sequencing reads obtained through whole-genome sequencing and report the 15,338 bp mitogenome of *Lerema accius*. The mitogenome is AT-rich and encodes 13 proteins, 22 transfer-RNAs, and two ribosomal-RNAs, with a gene order typical for Lepidoptera mitogenomes. A phylogenetic study based on the protein sequences using both Bayesian Inference and Maximum Likelihood methods consistently place *Lerema accius* with other grass skippers (Hesperiinae).

## INTRODUCTION

The order Lepidoptera contains approximately 160,000 described and half a million estimated species (*Kristensen, Scoble & Karsholt, 2007*). It represents one of the most diverse and fascinating groups of insects with many species emerging as model organisms for genetics and evolution (*Clarke & Sheppard, 1972*; *Nishikawa et al., 2013*; *Kunte et al., 2014*; *Zhan et al., 2011*; *Hines et al., 2012*; *Surridge et al., 2011*; *Engsontia et al., 2014*; *Zhang, Kunte & Kronforst, 2013*). Studies of these model organisms benefit significantly from decoding the genomes of select Lepidoptera species. Recently, we published the genome draft of Clouded Skipper *Lerema accius* using next generation sequencing techniques (*Cong et al., 2015*). Traditional genome assemblers failed to automatically assemble the Clouded Skipper mitogenome together with the nuclear genome. This failure probably resulted from a difficulty in distinguishing the mitogenome NGS reads from those of nuclear genome as well as a high erroneous $k$-mers frequency due to high mitochondrial DNA coverage. However, a dedicated effort should allow assembly of the mitogenome from whole-genome sequencing reads.

The insect mitogenome is circular, consisting of 14–19 kilobases (kb) that contain 13 protein-coding genes (PCGs), two ribosomal-RNA-coding genes (rRNAs), 22 transfer-RNA-coding genes (tRNAs), and an A + T rich displacement loop (D-loop) control region (*Cameron, 2014*). Because of their maternal inheritance, compact structure, lack of

Corresponding author
Nick V. Grishin,
grishin@chop.swmed.edu

genetic recombination, and relatively fast evolutionary rate, mitogenomes have been used widely in molecular phylogenetics and evolution studies (*Cameron, 2014*; *Moritz, Dowling & Brown, 1987*). Here, we assemble and annotate the complete mitogenome of *Lerema accius* from next generation sequencing reads. Phylogenetic analyses using published mitogenomes of skipper butterflies (Hesperiidae) place *Lerema accius* among other grass skippers (Hesperiinae).

## METHODS

### Library preparation and Illumina sequencing

We collected a male *Lerema accius* adult in the field (USA: Texas: Dallas County, Dallas, White Rock Lake, Olive Shapiro Park, 10-Nov-2013, GPS: 32.8621, −96.7305, elevation: 141 m) under permit #08-02Rev from Texas Parks and Wildlife Department (Natural Resources Program Director David H. Riskind). We removed the wings and abdomen of the deceased specimen (USA: Texas: Dallas County, Dallas, White Rock Lake, Olive Shapiro Park, 10-Nov-2013, GPS: 32.8621, −96.7305, elevation: 141 m), and used the remaining tissue to extract genomic DNA using the ChargeSwitch gDNA mini tissue kit (Life Technologies, Grand Island, NY, USA). About 500 ng of genomic DNA was used to prepare 250 bp and 500 bp paired-end libraries, respectively, following the Illumina TruSeq DNA sample preparation guide using enzymes from NEBNext Modules (New England Biolabs, Ipswich, MA, USA). These two libraries were pooled (and they occupied about 60% of one illumina lane) together with other libraries (not used for the mitogenome assembly) to sequence 150 bp from both ends with a rapid run on the Illumina HiSeq 2500 platform at the UT Southwestern Medical Center genomics core facility. The sequencing reads have been deposited in NCBI SRA database under accession numbers: SRR2089773– SRR2089775.

### Mitogenome assembly

Sequencing reads were processed by MIRABAIT (*Chevreux, Wetter & Suhai, 1999*) to remove contamination from sequence adapters and trimmed low-quality regions (quality score <20) at both ends. Using the mitogenomes of four skippers (*Carterocephalus silvicola*, *Potanthus flavus*, *Polytremis nascens* and *Polytremis jigongi*) as references, we applied mitochondrial baiting and iterative mapping (MITObim) v1.6 (*Hahn, Bachmann & Chevreux, 2013*) to extract the sequencing reads of the mitogenome in the 250 bp and 500 bp libraries.

About 1,161,000 reads (1.04% of all reads) were extracted using MITObim. Because the average size of the reference mitogenomes is 15,400 bp, we expected an average coverage of about 22,600 fold (1,161,000×150×2/15,400). We used JELLYFISH software (*Marcais & Kingsford, 2011*) to obtain the frequencies of 15-mers in these reads. The frequencies of some 15-mers were much lower than expected. They might come from regions in the mitogenome that were poorly covered in the sequencing reads; alternately, they might arise from sequencing errors, heterogeneity in different copies of mitochondrial DNA and reads from the nuclear genome. The second scenario could cause problems in *de novo* assembly, and thus we applied QUAKE (*Kelley, Schatz & Salzberg, 2010*) to correct

errors in 15-mers with frequencies lower than 1,000 and excluded reads containing low-frequency 15-mers after error correction. We assembled the error-corrected reads into contigs *de novo* with Platanus (*Kajitani et al., 2014*). The contigs were further assembled into scaffolds using all the reads (including the ones containing 15-mers with frequencies lower than 1,000).

This automatic procedure assembled a draft mitogenome of 15,332 bp without any gaps. However, since the genome assembler Platanus is not deigned to assemble circular mitogenomes, the linear representation of the circular DNA may either (1) miss a fragment after its 3′-terminus and before its 5′-terminus or (2) have redundant fragments that appear both at the 3′-terminus and 5′-terminus. We manually inspected the sequences at the 5′- and 3′-termini and revealed that there was no redundant fragment but instead a fragment of six base pairs was missing. We determined the sequence of the missing fragment by searching for the two 32 bp fragments at the 5′- and 3′-termini of the draft mitogenome in the sequencing reads and selected the sequence between them. A majority (99.8%) of the reads revealed the same missing fragment (others likely contained sequencing errors) and we manually added it into the mitogenome. We also adjusted the linear representation of the circular DNA by circular permutation so that the sequence started with the trnM(cau) gene, which was the convention for most Lepidoptera sequences deposited in the database.

## Annotation and analysis of the mitochondrial genome

The mitogenome sequence was annotated using the MITOS web server (*Bernt et al., 2013*). We translated the sequences of PCGs to protein sequences using the genetic code for invertebrate mitogenomes. The predictions from MITOS were manually curated using other published skipper mitogenomes as references, and the starts and ends of genes were modified, if necessary, to be consistent with other species. The open reading frames (after modification) of the protein coding genes were validated. Secondary structures of tRNA genes were predicted using the same server.

## Assembly quality assessment

We mapped the 250 bp and 500 bp paired-end reads to the mitogenome using bowtie2 v2.2.3 (*Langmead & Salzberg, 2012*) and processed the results with SAMtools (*Li et al., 2009*). Coverage depth at each position was calculated based on this mapping result. As the sequencing reads that could map partly to the 5′-terminus and partly to the 3′-terminus would map only to one terminus or fail to map, the coverage at the termini could be under-estimated. Therefore, we recalculated the coverage for the 1,000 bp segments in the 5′- and 3′-termini based on the mapping result to another linear representation of the circular mitogenome that was obtained by connecting the 5′-terminal half to the end of 3′-terminal half.

Only two regions of the mitogenome showed coverage below 1,000 fold. One of them was a low complexity region that contained mostly (85%) T and another was a 46 bp fragment of AT repeats. Such AT-rich regions tend to be underrepresented in the sequencing libraries as they break easier during the library preparation (*Benjamini & Speed, 2012*).

**Table 1  List of taxa analyzed in present paper.**

| Species | Length | Identity[a] | Accession | References |
|---|---|---|---|---|
| *Ampittia dioscorides* | 15,313 | 91.2% | KM102732.1 | XW Yang et al., 2014, unpublished data |
| *Apocheima cinerarium* | 15,722 | n.a. | NC_024824.1 | *Liu et al. (2014)* |
| *Biston suppressaria* | 15,628 | 99.7% | NC_027111.1 | *Chen et al. (2015)* |
| *Carterocephalus silvicola* | 15,765 | 99.0% | NC_024646.1 | *Kim et al. (2014)* |
| *Celaenorrhinus maculosa* | 15,282 | n.a. | NC_022853.1 | *Wang, Hao & Zhao (2013)* |
| *Choaspes benjaminii* | 15,300 | 85.9% | NC_024647.1 | *Kim et al. (2014)* |
| *Ctenoptilum vasava* | 15,468 | n.a. | NC_016704.1 | *Hao et al. (2012)* |
| *Daimio tethys* | 15,350 | 96.8% | NC_024648.1 | *Kim et al. (2014)* |
| *Erynnis montanus* | 15,530 | 99.8% | NC_021427.1 | *Wang et al. (2014)* |
| *Graphium timur* | 15,226 | 97.9% | NC_024098.1 | *Chen et al., (2014)* |
| *Hasora anura* | 15,290 | n.a. | NC_027263.1 | *Wang et al. (2015)* |
| *Lobocla bifasciata* | 15,366 | 95.9% | NC_024649.1 | *Kim et al. (2014)* |
| *Ochlodes venata* | 15,622 | 78.5% | NC_018048.1 | C Xu et al., 2012, unpublished data |
| *Papilio glaucus* | 15,306 | 99.5% | NC_027252.1 | *Shen, Cong & Grishin (2015)* and *Cong et al., (2015)* |
| *Parnassius apollo* | 15,404 | 98.6% | NC_024727.1 | *Kim et al. (2009)* |
| *Phthonandria atrilineata* | 15,499 | 99.9% | NC_010522.1 | *Yang et al. (2009)* |
| *Polytremis jigongi* | 15,353 | 99.6% | NC_026990.1 | *Jiang et al. (2015)* |
| *Polytremis nascens* | 15,392 | 83.6% | NC_026228.1 | *Jiang et al. (2015)* |
| *Potanthus flavus* | 15,267 | 99.3% | NC_024650.1 | *Kim et al., (2014)* |

Notes.

[a]Identity: the lowest sequence identity to independently sequenced mitochondrial DNA of the same species in the Non-redundant database identified by BLAST.

n.a.: there is no other mitochondrial sequences of the same species in the Genbank for cross-validation.

Manual inspection by searching the flanking regions of these poorly covered fragments in the sequencing reads revealed variation in the length of the poly-T sequence in the first fragment and the number AT-repeats in the second fragment, respectively. The variation might correspond to the heterogeneity in different copies of mitochondrial DNA in the specimen. We confirmed that the mitogenome produced by the de novo assembler did represent the dominant form of the possible variations.

We further assessed the quality of our assembly by its consistency with other published skipper mitogenomes in the protein-, rRNA- and tRNA-coding regions. We aligned the rRNA- and tRNA-coding sequences directly and aligned translated sequences for PCGs. Alignments confirmed that our sequences were consistent with the majority of available mitogenomes, and gaps were only in regions that are poorly conserved among other skipper species. In addition, the COI barcode (5′-terminal region of cytochrome oxidase subunit 1 coding gene) of *Lerema accius* was reported previously (Genbank accession: GU088418.1) and this sequence agreed 100% with the corresponding region in our mitogenome.

## Phylogenetic analysis

The mitogenomes of 13 other skipper species that were available (up to June, 2015) were downloaded from NCBI (Table 1). Three moths from the Geometridae family and three species of the Papilionidae family were used as outgroups. A blast search against all the available sequences of the same species in the non-redundant database was used to validate

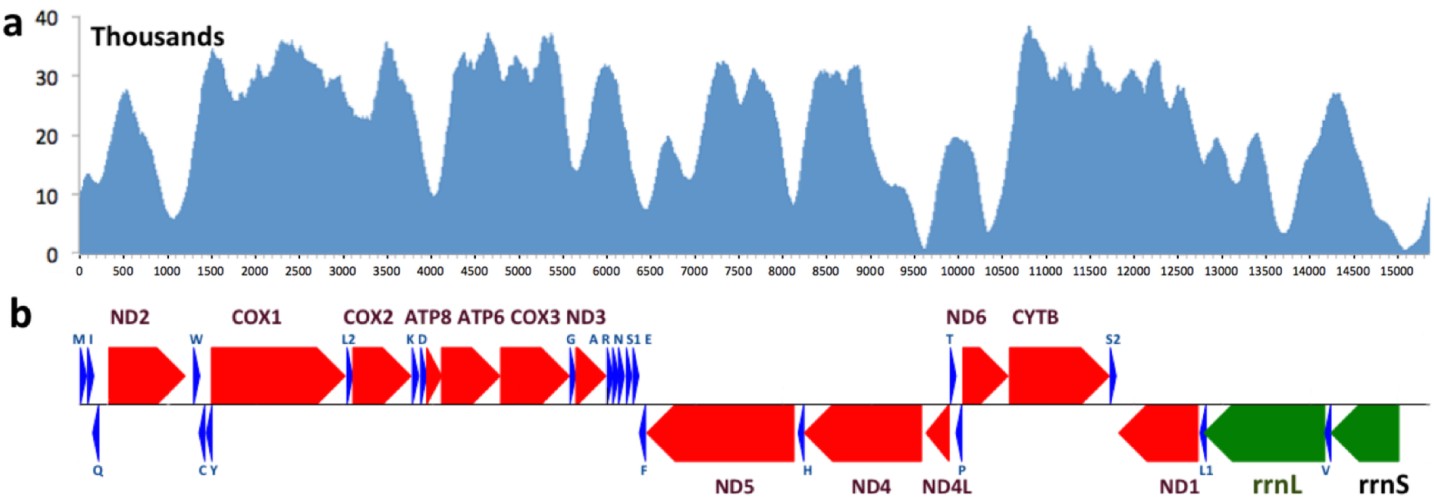

**Figure 1** **Coverage and annotation of *Lerema accius* mitogenome.** The same base positions are aligned between (A) and (B). (A) Coverage by sequencing reads at each base position. (B) Map of genes in the *Lerema accius* mitogenome. PCGs are colored in red, tRNA-coding genes are in blue, rrnL and rrnS are in green. Each gene is shown as an arrow indicating the transcription direction. The arrows on top of the black line correspond to genes coded on the majority strand, and those below show genes on the minority strand.

each mitogenome sequence. For most species, some individual genes in the mitogenome were sequenced independently and the mitogenome sequence was consistent with these gene sequences (sequence identity >95%). However, four skippers were excluded from downstream analyses (*Ampittia dioscorides, Choaspes benjaminii, Ochlodes venata* and *Polytremis nascens*, three of which are unpublished but available from GenBank) due to poor agreement for at least one gene sequence found in GenBank.

Protein sequences of the 13 protein-coding genes were aligned by MAFFT. We manually checked the alignments, corrected annotation errors based on consensus and removed positions with long gaps and their surrounding regions with uncertain alignment. The processed alignments were concatenated and analyzed with Bayesian Inference and Maximum likelihood methods using Phylobayes-MPI v1.5a (*Lartillot, Lepage & Blanquart, 2009*) (model: CATGTR (*Lartillot & Philippe, 2004*)) and RaxML v8.1.17 (*Stamatakis, 2014*) (model: PROTGAMMAAUTO), respectively. The resulting phylogenetic trees were visualized in FigTree v1.4.2.

## RESULTS AND DISCUSSION

### Annotation of the mitogenome

The complete mitogenome of *Lerema accius* is deposited in GenBank of NCBI under accession number KT598278. The length of this mitogenome is 15,338 bp and it retains the typical insect mitogenome gene set and gene order, including 13 PCGs (nd1-6, nd4l, cox1-3, atp8, atp6, and cytb), 22 tRNA genes (two for Serine and Leucine and one for each of the rest of the amino acids), 2 ribosomal RNAs (rrnL and rrnS), and an A + T rich D-loop control region. The annotation of the mitogenome is illustrated in Fig. 1. The cox1 gene uses start codon CGA, which is consistent with many other insect mitogenomes (*Kim et al., 2009*). All the rest of the genes start with the typical ATN. cox1, cox2 and nd4

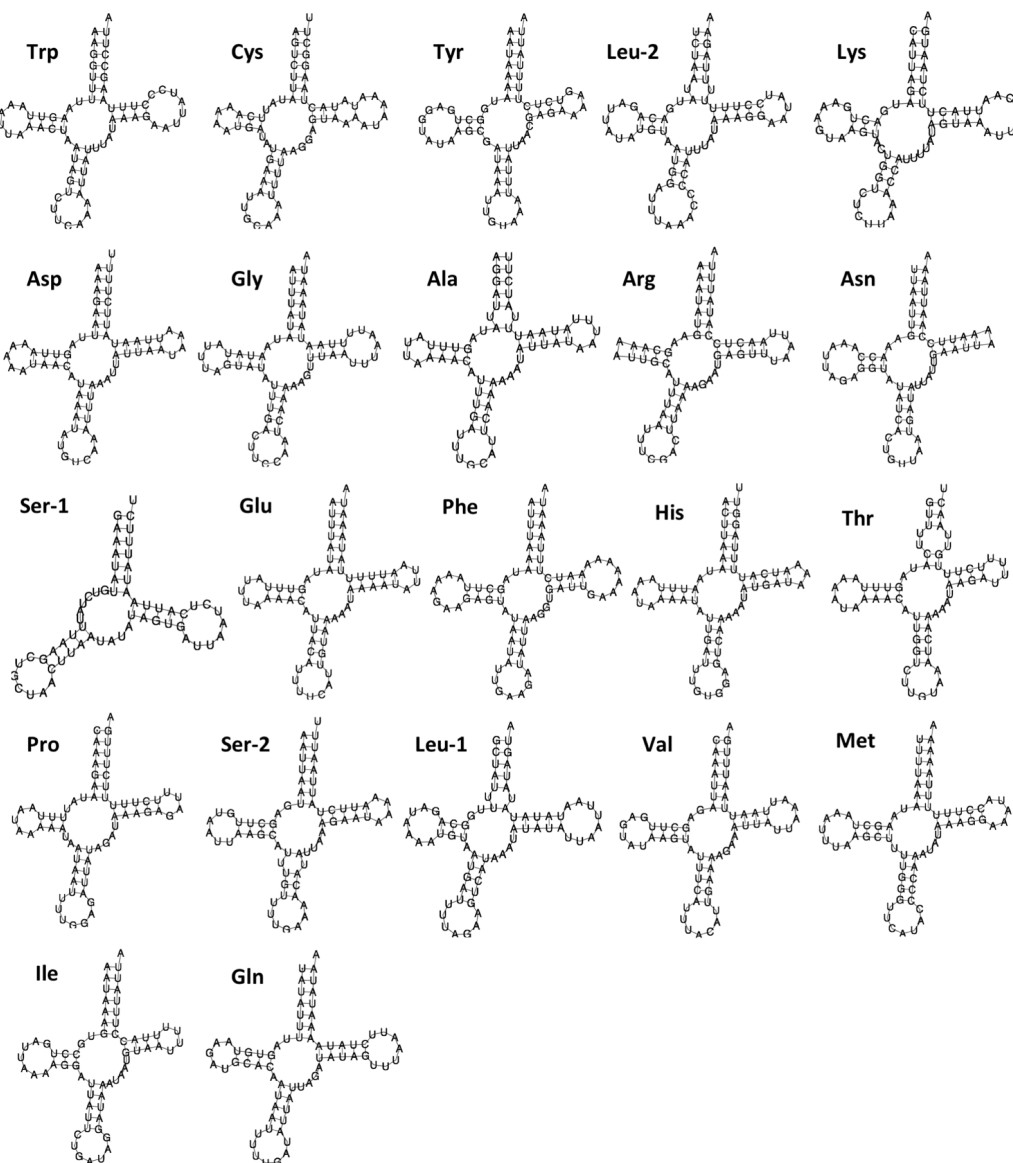

**Figure 2 Secondary structure of 22 tRNAs encoded by the *Lerema accius* mitogenome.** The tRNAs are labeled by the abbreviations of their corresponding amino acids.

use an incomplete stop codon T (*Ojala, Montoya and Attardi, 1981*), and a complete TAA codon will likely be formed during mRNA maturation (*Ojala, Montoya and Attardi, 1981*; *Boore, 1999*).

The lengths of tRNA-coding genes range from 60 bp to 70 bp. Secondary structures predicted by MITOS suggest that all tRNAs adopt a typical cloverleaf structure except for trnS1(gcu) (Fig. 2). The dihydrouridine (DHU) arm of trnS1(gcu) does not form a stable stem-loop structure, which is very common in butterfly mitogenomes (*Lu et al., 2013*; *Kim et al., 2014*). A 488 bp A + T rich region (A + T content: 94.7%) connects rrnS and trnM(cau). This region contains an "ATAGA" motif located 22 bp downstream from rrnS
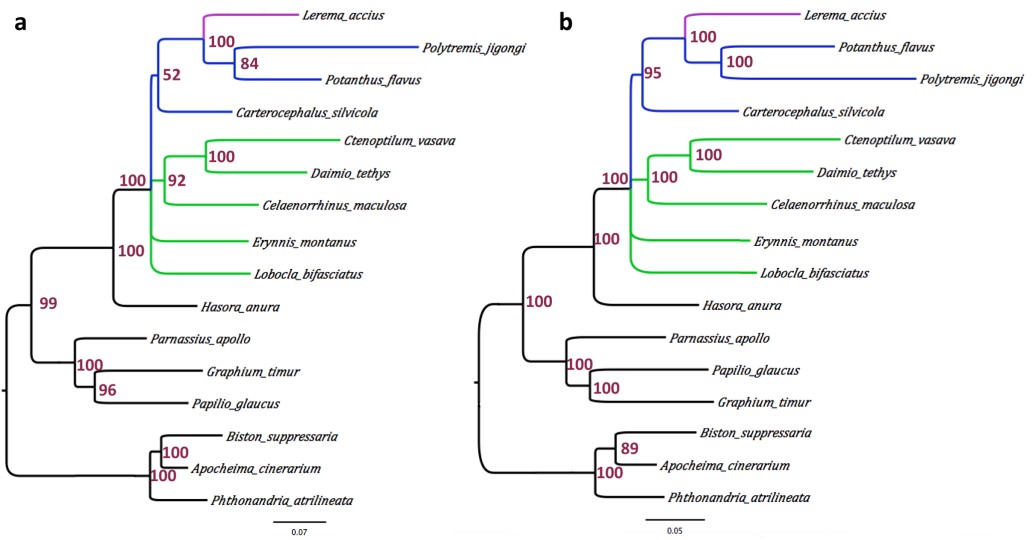

**Figure 3** **Phylogeny of skippers based on the concatenated alignment of the mitochondrial protein sequences.** (A) Consensus of phylogenetic trees by RAxML (MTZOA model) based on bootstrap samples of the alignment. (B) Consensus of phylogenetic trees sampled by Phylobayes-MPI (v1.5a) with CATGTR model.

and is followed by 15 bp of poly-T stretch that is a gene regulation element commonly found in Lepidoptera (*Lu et al., 2013*; *Salvato et al., 2008*).

## Phylogenetic analysis

We built a phylogenetic tree of the 10 skipper species with published mitogenomes, based on the concatenated alignment of the mitochondrial protein sequences. Three Papilionidae and three Geometridae mitogenomes were used as outgroups. Maximum likelihood method RAxML automatically selected MTZOA, a general mitochondrial amino acid substitution model, as the most appropriate, and placed *Lerema accius* among other grass-skippers (Subfamily Hesperiinae). A Bayesian analysis with the CATGTR model supported a tree with exactly the same topology. This topology is largely consistent with previously reported phylogenetic studies on the basis of standard gene markers and morphology (*Warren, Ogawa and Brower, 2008*; *Warren, Ogawa and Brower, 2009*; *Yuan et al., 2015*). Notably, the subfamily Coeliadinae (represented by *Hasora anura*) is a sister to all other Hesperiidae. Topology between the subfamilies Eudaminae (*Lobocla bifasiatus*), Pyrginae (other branches shown in green in Fig. 3) and remaining Skippers is unresolved. The tribes Celaenorrhini (*Celaenorrhinus maculosa*) and Tagiadini (*Daimio* and *Ctenoptilum*) group together (in the absence of Pyrrhopygini). The subfamily Heteropterinae (represented by *Carterocephalus silvicola*) is a sister to grass skippers (Hesperiinae).

Interestingly, based on the mitochondrial genome, the two Asian grass skippers (*Potanthus* from the tribe Taractrocerini and *Polytremis* from the tribe Baorini) are grouped together, and *Lerema* (from the tribe Moncini) is their sister. The sequences of two nuclear markers, EF1a and wingless, are available from these species in the database; however, they support different topologies at low confidence. While the maximal likelihood

tree based on EF1a favors (bootstrap: 52%) the same topology as the mitogenome, the tree based on wingless groups *Potanthus* with Lerema with a 63% bootstrap support and places *Polytremis* as their sister. The phylogeny between these tribes could become clear when more sequences from more taxa become available.

### Funding

This work was supported by the National Institutes of Health (GM094575 to NVG) and the Welch Foundation (I-1505 to NVG). Qian Cong is a Howard Hughes Medical Institute International Student Research fellow. The funders had no role in study design, data collection and analysis, decision to publish, or preparation of the manuscript.

### Grant Disclosures

The following grant information was disclosed by the authors:
National Institutes of Health: GM094575.
Welch Foundation: I-1505.
Howard Hughes Medical Institute International.

### Competing Interests

The authors declare there are no competing interests.

### Author Contributions

- Qian Cong conceived and designed the experiments, performed the experiments, analyzed the data, wrote the paper, prepared figures and/or tables, reviewed drafts of the paper.
- Nick V. Grishin conceived and designed the experiments, contributed reagents/materials/analysis tools, wrote the paper, reviewed drafts of the paper.

### Field Study Permissions

The following information was supplied relating to field study approvals (i.e., approving body and any reference numbers):

Texas Parks and Wildlife Department (Natural Resources Program Director David H. Riskind) issued permit #08–02Rev for Texas State Parks.

### DNA Deposition

The following information was supplied regarding the deposition of DNA sequences:

The complete mitogenome of Lerema accius is deposited in GenBank of NCBI under accession number KT598278.

### Data Availability

The sequencing reads have been deposited in NCBI SRA database under accession numbers: SRR2089773– SRR2089775.

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
