# Peer review of "The complete mitochondrial genome of Lerema accius and its phylogenetic implications"

_PeerJ, doi:10.7717/peerj.1546_

## Round 0.1 · original submission · Minor Revisions

I now have reviews back from 2 referees who are both enthusiastic about the manuscript and agree that only minor revisions are needed to make the manuscript suitable for publication. In revisions, I ask that you pay particular attention to the request for additional details in the methods and the naming conventions for mitochondrial genes pointed out by the referees. Still, I expect that it will be relatively straightforward for you to complete these revisions, and I look forward to seeing your revised manuscript.

Reviewer 1 ·

Basic reporting

The authors meet standards for basic reporting.

With the emergence of butterflies and moths as model organisms, the authors wished to assemble the mitochondrial genome for Lerema accius, for which a nuclear genome was previously assembled by the same group. The authors have presented a rather straightforward process in which whole genome DNA from a single individual was prepared using standard techniques and sequenced on the Illumina HiSeq platform. Mitochondrial baiting and iterative mapping using several published reference mitogenomes were used to extract sequencing reads from the complete set of reads. A de novo assembly was performed on this subset of reads, and the resulting contig was circularized, evaluated for consistency with other published genomes, and annotated. Phylogenetic analyses with other published mitogenomes yielded largely expected results, with one unexpected topology.

Experimental design

The authors sequence a single specimen of the species, and use standard methods to assemble the mitogenome.

Validity of the findings

The manuscript is very straightforward, and the authors appear to have followed due diligence in constructing the mitochondrial genome, although neither raw data, nor NCBI accession numbers were provided for peer review. Recommended edits to improve ease of reading may be found on the “track edits” version of the reviewed manuscript.

Additional comments

Keywords are all found in the title. Choose keywords that will enhance searches beyond what your title provides.

Methods: This paper is lacking a significant amount of detail regarding specimen collection and preparation of reads included in the Cong et al. 2015 paper. If the same methods were used to produce sequences as in the Cong et al. 2015 paper, these should be included in this paper as well. For example, it is not clear from the methods section in this paper that DNA was prepared using multiple kits, then pooled prior to sequencing, as was explained in the Cong et al. 2015 paper. In this paper, it appears they were sequenced separately, and processed separately downstream.

Conclusions: The authors mention an unexpected topology in which “the two Asian grass skippers (Potanthus from the tribe Taractrocerini and Polytremis from the tribe Baorini) are grouped together, and Lerema (from the tribe Moncini) is their sister.” As this is a follow-up study to a whole-genome sequencing effort in which the nuclear genome was assembled (Cong et al. 2015), was this unexpected result consistent with previous results based on nuclear markers? Are barcodes, such as COI, which is stated in the Cong et al. 2015 paper as being very useful for this taxon, available that could be used to verify this result?

Table 2 is somewhat redundant to Figure 1. If bp are added to Figure 1 so gene locations within the mitogenome may be discerned from the figure, Table 2 may be moved to supplemental materials.

Annotated reviews are not available for download in order to protect the identity of reviewers who chose to remain anonymous.

·

Basic reporting

The manuscript characters the complete Mitogenomes Lerema accius by Next Generation Sequencing methods. Although there are no different to the structure of other Lepidoptera, the sequence method is novelty. I suggest it can be accept after minor revision.
1 part ‘Mitogenome assembly’, ‘after its C--‐terminus and before its N--‐terminus’, to DNA, we always used 5’- or 3’ terminus; to protein, C--‐terminus or N--‐terminus always be used. Please check it is OK?
2 Part ‘Annotation of the mitogenome’, Genbank should be GenBank, Why number only XYZ? Please check it.
3 In general, there is not complete agreement on the correct designation and naming of the mitochondrial genes. However, the most "standardized" convention (i.e. see Boore LJ: Requirements and standards for organelle genome databases. OMICS A journal of Integrative Biology 2006, 10(2):119-126.) suggests to use lowercase and italic letters when referring to PCGs (i.e. atp8, not ATP8) and lowercase and italic letter with a single letter in capitals when referring to a tRNA gene (i.e. trnL(uaa) not tRNA-Leu).
4 When writing numbers in regular text, generally those under ten are written out, while those ten and above can be in the form of numerals. Make sure this is consistent throughout the paper.

Experimental design

Well shown.

Validity of the findings

Well shown.

---

## Round 0.2 · accepted · Accept

I am satisfied that you have addressed the comments of the reviewers in your revision, and am happy to move your paper forward into publication. I note that there are still some issues with the English in the manuscript (e.g., "about 60% of a illumine lane" should read "about 60% of an Illumina lane") but beyond these minor corrections, the paper is sound and ready for acceptance.